# Epidemiology of leptospirosis in Tanzania: A review of the current status, serogroup diversity and reservoirs

Shabani Kiyabo Motto[1,2]*, Gabriel Mkilema Shirima[1], Barend Mark de Clare Bronsvoort[3,4], Elizabeth Anne Jessie Cook[5,6]*

1 Department of Global Health and Bio-Medical Sciences, School of Life Science and Bio-engineering, The Nelson Mandela African Institution of Science and Technology, Arusha, Tanzania, 2 Tanzania Veterinary Laboratory Agency, Central Veterinary Laboratory, Dar es Salaam, Tanzania, 3 The Roslin Institute, University of Edinburgh, Easter Bush, United Kingdom, 4 Centre for Tropical Livestock Genetics and Health, The Roslin Institute, University of Edinburgh, Easter Bush, United Kingdom, 5 International Livestock Research Institute (ILRI), Nairobi, Kenya, 6 Centre for Tropical Livestock Genetics and Health, ILRI, Nairobi, Kenya

* skymotto@gmail.com (SKM); e.cook@cgiar.org (EAJC)

**Data Availability Statement:** All relevant data are presented within the manuscript and its Supporting Information files.

## Abstract

### Background

Tanzania is among the tropical countries of Sub-Saharan Africa with the environmental conditions favorable for transmission of *Leptospira*. Leptospirosis is a neglected zoonotic disease, and although there are several published reports from Tanzania, the epidemiology, genetic diversity of *Leptospira* and its host range are poorly understood.

### Methods

We conducted a comprehensive review of human and animal leptospirosis within the 26 regions of the Tanzanian mainland. Literature searches for the review were conducted in PubMed and Google Scholar. We further manually identified studies from reference lists among retrieved studies from the preliminary search.

### Results

We identified thirty-four studies describing leptospirosis in humans (n = 16), animals (n = 14) and in both (n = 4). The number of studies varied significantly across regions. Most of the studies were conducted in Morogoro (n = 16) followed by Kilimanjaro (n = 9) and Tanga (n = 5). There were a range of study designs with cross-sectional prevalence studies (n = 18), studies on leptospirosis in febrile patients (n = 13), a case control study in cattle (n = 1) and studies identifying novel serovars (n = 2). The most utilized diagnostic tool was the microscopic agglutination test (MAT) which detected antibodies to 17 *Leptospira* serogroups in humans and animals. The *Leptospira* serogroups with the most diverse hosts were Icterohaemorrhagiae (n = 11), Grippotyphosa (n = 10), Sejroe (n = 10), Pomona (n = 9) and Ballum (n = 8). The reported prevalence of *Leptospira* antibodies in humans ranged from 0.3–

**Funding:** This research was conducted as part of the CGIAR Research Program on Livestock. ILRI is supported by contributors to the CGIAR Trust Fund. CGIAR is a global research partnership for a food-secure future. Its science is carried out by 15 Research Centers in close collaboration with hundreds of partners across the globe (www.cgiar. org). This research was funded in part by the Bill & Melinda Gates Foundation and with UK aid from the UK Foreign, Commonwealth and Development Office (Grant Agreement OPP1127286) under the auspices of the Centre for Tropical Livestock Genetics and Health (CTLGH), established jointly by the University of Edinburgh, SRUC (Scotland's Rural College), and the International Livestock Research Institute (ILRI). This work was also supported by funding from the BBSRC (BBS/E/D/ 30002275). The funders had no role in study design, data collection and analysis, decision to publish, or preparation of the manuscript.

**Competing interests:** The authors have declared that no competing interests exist.

29.9% and risk factors were associated with occupational animal contact. Many potential reservoir hosts were identified with the most common being rodents and cattle.

## Conclusion

Leptospirosis is prevalent in humans and animals in Tanzania, although there is regional and host variation in the reports. Many regions do not have information about the disease in either humans or their animal reservoirs. More studies are required to understand human leptospirosis determinants and the role of livestock in leptospirosis transmission to humans for the development of appropriate control strategies.

## Author summary

Bacteria from the genus *Leptospira* is an important agent for causing a disease called leptospirosis in humans and a range of animal species. Leptospirosis is often under-recognized as it presents varied symptoms that mimic malaria, typhoid, brucellosis and other diseases. More than 250 pathogenic *Leptospira* serovars are known to cause leptospirosis in humans and animals. The diversity of *Leptospira* serovars and their distribution in humans and animals is little defined in Tanzania. We conducted a systematic review to gather information on the diversity of *Leptospira* serovars with their reservoir distribution and the most common diagnostics methods used. We included studies (n = 34) in the review and found 17 serogroups described in 28 studies that utilized microscopic agglutination test (MAT). So far human and other animal hosts including cattle, dogs, pigs, bats, buffalo, fish, rodents, goats, lion, zebra, sheep and shrews have been investigated for leptospirosis in Tanzania. Our results show that cattle and rodents are likely to be important reservoirs of pathogenic *Leptospira* spp. and can be a source of human leptospirosis principally in the farming system. Further studies are needed to explore predominant serovars in livestock for the development of prevention strategies to reduce transmission and risks in humans.

## Introduction

Leptospirosis is a serious infectious disease caused by spirochete bacteria in the genus *Leptospira* [1]. It is considered a re-emerging zoonosis widespread in tropical and sub-tropical regions, where there are limited surveillance and disease control measures [2]. Leptospirosis infections may be acute, subacute or chronic [1] and may result in severe health problems such as pulmonary haemorrhagic syndrome (PHS) [3], or renal and liver dysfunctions [2,4,5]. Leptospirosis often presents with varied symptoms that mimic those of several other unrelated febrile illnesses including dengue and malaria [6]. Therefore, leptospirosis is an important undifferentiated febrile illness that requires differential diagnosis [7].

The incidence of leptospirosis is poorly known and this may be partially attributed to inadequate data and surveillance [8]. In addition, there is a shortage of appropriate diagnostic facilities in developing countries, and clinicians may fail to recognize leptospirosis in febrile patients, consequently it remains underreported [2]. However, it is estimated that around the globe there are 1.03 million leptospirosis cases annually and 2.9 million Disability Adjusted Life Years (DALYs), where the majority of infections and burden are in low and middle-income countries (LMICs) [4,9].

Leptospires are mainly harboured in the renal tubule and excreted in the urine of accidental and maintenance hosts including cattle, rodents, pigs, dogs, sheep and goats [1,10]. Humans contract leptospirosis from contaminated environments, consumption or handling waste products from infected animals [1,11]. More than 250 serovars have been serotyped into 31 serogroups which can potentially cause leptospirosis in humans and animals worldwide [12,13]. Based on DNA hybridization techniques and phylogenetic analysis, 64 species have been recognized and rearranged into two clades (pathogenic "P" and saprophytic "S") and two subclades in each clade (subclade P1 and P2 and subclades S1 and S2) [14]. There are 17 species classified in subclade P1 of which 8 can cause severe disease in humans and 21 species in subclade P2 that can cause mild disease, and the remaining species, considered non-pathogenic, are in clade S subclade S1 and S2 [14].

Leptospirosis in Tanzania was reported in the early 1990s [15]. The authors of that study aimed to determine seroprevalence in humans, domestic and wild animals based on the microscopic agglutination test (MAT). The seroprevalence of *Leptospira* antibodies was reported as 38% in dogs, 5.6% in cattle, 1.8% in rodents and 0.3% in humans [15]. Despite the low prevalence of *Leptospira* antibodies in humans, it was sufficient to indicate a public health concern and the need for control and prevention strategies. Several studies have been conducted since and leptospirosis has been reported in a range of species [11,16–18]. Two recent investigations estimated human leptospirosis incidence in Tanzania. The study populations involved were hospitalized patients with fever related symptoms. The disease incidence was estimated by the two studies to be 75-102/100,000 persons annually in 2007–2008 [19] and 11-18/100,000 persons annually in 2012–2014 [20]. Humans are at high risk of contracting leptospirosis based on the fact that multiple animal species harbour and transmit the disease including livestock and wildlife [11,18,21]. Although three decades have elapsed since the first detection of leptospirosis in Tanzania the epidemiology and the diversity of leptospiral serovars and their reservoirs are not well articulated. This review comprehensively examined the disease epidemiology and *Leptospira* diversity in Tanzania to inform stakeholders of any existing knowledge gaps and for appropriate management of the disease.

## Methods

### Search strategy

A thorough and comprehensive search of the literature was carried out to identify studies associated with human, domestic or wild animal leptospirosis and *Leptospira* in Tanzania. To retrieve all related information, a boolean operator ("OR" and "AND") with a combination of keywords was set and both PubMed and Google Scholar electronic search engines were used to retrieve published papers, peer-reviewed articles, theses, case reports, posters and conference presentations. Retrieval of materials from PubMed and Google search engine was done on 24th May 2020. In the PubMed search engine search terms were: ('human' OR 'people' OR 'domestic animals' OR 'bovine' OR 'cattle' OR 'pigs' OR 'porcine' OR 'rodent' OR 'rat' OR 'dogs' OR 'canine' OR wildlife' OR 'wild animals') AND ("leptospirosis" OR '*Leptospira*' OR 'Weils disease' OR 'Weils syndrome' OR '*Leptospira* serovars' OR 'sokoine serovar' OR 'interrogans serovar' OR 'Icterohaemorrhagiae serovar' OR 'Hebdomadis serovar') AND ('Tanzania' OR 'Northern zone' OR 'Kilimanjaro' OR 'Morogoro' OR 'Rukwa' OR 'Katavi' OR 'Tanga' OR 'Kagera' OR 'Simiyu' OR 'Mara' OR 'Geita' OR 'Shinyanga' OR 'Songwe' OR 'Moshi') AND ('prevalence' OR 'epidemiology' OR 'risk factors' OR 'febrile illness' OR 'acute leptospirosis'); while in Google scholar (('*Leptospira*' OR 'leptospirosis') AND Tanzania)) were the key search terms used.

## Study selection

The search returned a large number of publications, and the contents were collated in Mendeley citation manager version 1.19.4. Additional papers were identified from reference lists of retrieved articles to find appropriate studies that might not have been identified during the preliminary search. All papers were checked for duplicates and removed in Mendeley software. In the subsequent stage, those papers remaining after cleaning were then screened dependent on their titles and relevant geographical study location. Consequently, the full content of those papers was further assessed as far as their significance and by considering the inclusion and exclusion criteria.

## Criteria for study eligibility

**Inclusion and exclusion criteria.** In this review, all publications including published papers, theses, poster or conference presentations were included if the source contained primary data citing leptospirosis/ febrile illness in humans, domestic or wild animals. Theses and poster presentations were excluded if the data had been published in another peer-review journal. All texts written in English and focused on Tanzania as the geographical area of attention were eligible.

## Results

At a preliminary search, a total of 3767 documents were retrieved from two database search engines and pooled into the Mendeley citation manager. Of those articles, 3720 were recovered from Google Scholar and 47 from PubMed. A further 13 papers were searched and added manually after being identified from reference lists among the retrieved articles to make 3780 papers in total. Then articles were checked for duplicates in Mendeley, 3465 articles remained and met the criteria for the initial stage of inclusion and exclusion after duplicate removal. The initial screening was based on the title of the article and relevant study location (i.e. Tanzania), 3395 articles were excluded in the review process due to failure to fulfil the inclusion criteria for the next stage of assessment. A large number of articles recovered from Google scholar were excluded as they did not report leptospirosis in Tanzania. These articles were detected by the search engine because Tanzania was mentioned in the text of the paper as the author had referenced a previous publication. The publications were most often reporting leptospirosis in another country. After the selected literature underwent full text screening, 32 published papers were identified with primary data describing *Leptospira* and leptospirosis from Tanzania in humans and various animal species. In addition, two papers were identified, which were published after the initial retrieval was conducted, and these have been included in the review [22,23]. The flow diagram Fig 1 describes the process of identifying studies for this review. A summary of each study is available in S1 Table including the year of research, study design, geographical location, target populations, diagnostics tests, and results for each study (n = 34).

Of the 34 studies identified, sixteen described *Leptospira* seropositivity or leptospirosis in humans, fourteen investigated animals and four focused on both humans and animals S1 Table. There was a range of study designs with more than fifty per cent of studies being prevalence studies (n = 18). Over thirty percent were targeted studies investigating *Leptospira* as a cause of illness in febrile patients (n = 13) or disease in animals (n = 1) and a small number identified novel serovars (n = 2).

## Geographic distribution of *Leptospira* studies

The Tanzanian mainland comprises 26 regions that are divided into 6 zones which are as follows: Lake Zone (Mwanza, Kagera, Shinyanga, Geita, Mara and Simiyu), Western Zone

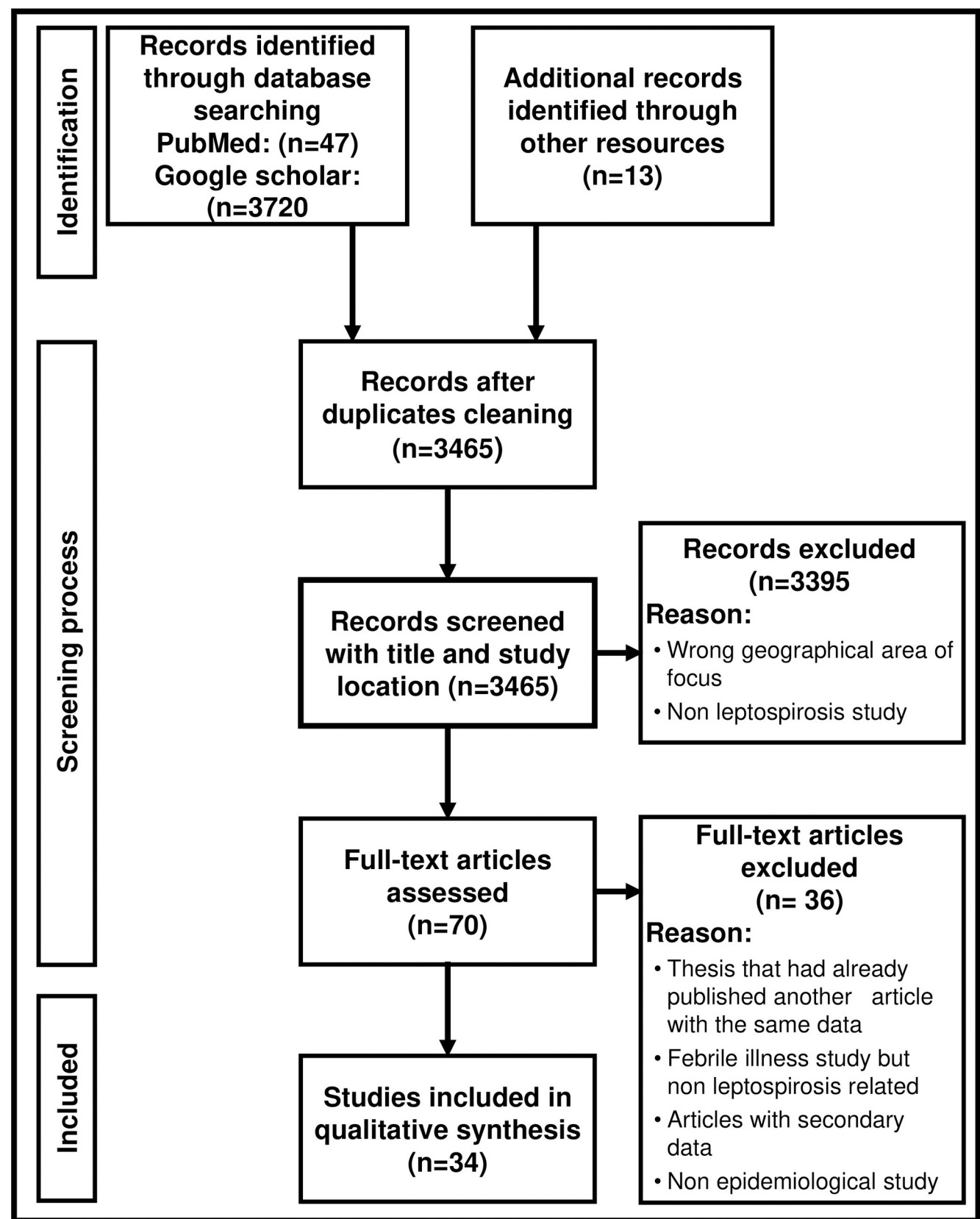

**Fig 1. Flow diagram indicating how articles were included in the review regarding leptospirosis in Tanzania.**

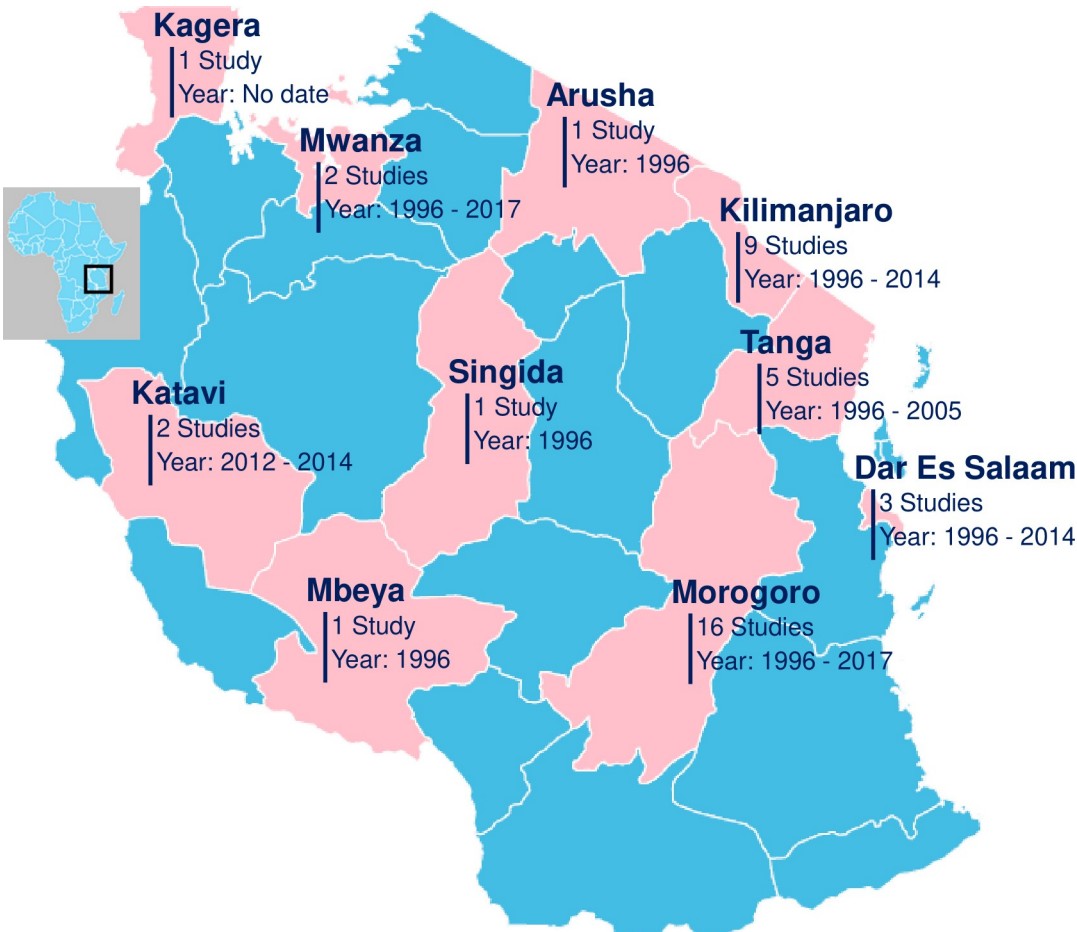

**Fig 2. Geographical distribution of *Leptospira* studies reported from human, domestic and wild animals: Regions colored pink indicate areas with *Leptospira* studies from 1990s to date and regions colored blue indicate regions where no study was retrieved from the search engine.** This map was prepared using Simplemaps https://simplemaps.com/resources/svg-tz.

(Katavi and Kigoma), Southern Highland Zone (Songwe, Rukwa, Ruvuma, Mbeya, Iringa and Njombe), Eastern Zone (Morogoro, Pwani, Dar es Salaam, Lindi and Mtwara), Central Zone (Dodoma, Singida and Tabora) and Northern Zone (Kilimanjaro, Manyara, Tanga and Arusha). The geographical distributions of the recovered studies are shown in Fig 2.

There was an unequal distribution in the *Leptospira* studies conducted across the country. Human or animal related studies were only conducted in 10 (38.5%) out of 26 regions of the Tanzanian mainland. The majority of the studies reporting *Leptospira* or leptospirosis were from Morogoro region (n = 16) followed by Kilimanjaro (n = 9) and Tanga (n = 5). Additional studies were conducted in Dar es Salaam (n = 3), Katavi (n = 2), Mwanza regions (n = 2), Kagera (n = 1), Arusha (n = 1), Singida (n = 1) and Mbeya (n = 1) Fig 2. Only one study was conducted in multiple regions [15]. In some regions such as Mbeya and Singida the research was conducted many years ago at the onset of the disease identification in the country.

Studies from Morogoro region (n = 16) were mostly cross-sectional studies in animals (n = 8), or humans and animals (n = 2) and among these the animals studied were: rodents, shrews, cattle, goats, sheep, pigs, dogs, cats, fish and bats. The other studies from Morogoro described leptospirosis in hospital patients (n = 4) or new serovars (n = 2). On the other hand, studies conducted in the Kilimanjaro (n = 9) region were hospital-based studies describing

leptospirosis in humans (n = 7). There was one cross-sectional study in humans and animals (n = 1) and one study focused only on animals with the target animals being cattle, goats, sheep and rodents. Among the five studies in Tanga, there were four cross-sectional studies, including two animal studies, one human study, one study in both humans and animals, and one targeted study investigated clinical disease in animals. Of the studies in Dar es Salaam two were hospital based and one was a cross sectional study of animals and humans. The study in Arusha was hospital based and the studies in the remaining regions were cross sectional in humans (Katavi and Mwanza) and in both humans and animals (Kagera, Mbeya, Katavi, Mwanza and Singida).

## Diagnostic approaches for detecting *Leptospira* or antibodies to *Leptospira*

Various diagnostic methods for leptospirosis were identified during the review S1 Table. These diagnostic techniques include microscopic agglutination test (MAT) (n = 28), culture and isolation (n = 7), cross agglutinin absorption test (CAAT)(n = 2), Eiken latex agglutination test (n = 1), enzyme linked immunosorbent assay (ELISA) (n = 1) and polymerase chain reaction (PCR) (n = 9). Despite the advancement of diagnostic technology, currently few studies use molecular typing [10,22,24] for characterising *Leptospira* sp. Most of the studies (n = 22) employed a single technique for leptospirosis detection. Microscopic agglutination test (MAT) was broadly utilized in 85% of the studies (n = 28) for leptospirosis diagnosis and in nine of these studies it was utilized in combination with other methods such as ELISA, culture, or PCR S1 Table. Recent studies used advanced diagnostics methods including either polymerase chain reaction, molecular typing or in combination (n = 9). For PCR, the studies used a variety of tissues such as kidney, culture isolate and blood sample for detection and the assays had different gene targets [10,22,23,25,26].

## Leptospiral serogroups used in studies that utilized MAT

The studies utilizing the MAT test for detection of antibodies to *Leptospira* included a wide range of *Leptospira* serogroups Fig 3. In general, human studies tended to use a wider range of serogroups compared to studies from animals Fig 3 [20,27–29]. Serogroups commonly used in human studies included: Australis (n = 7), Ballum (n = 9), Canicola (n = 4), Grippotyphosa (n = 9), Hebdomadis (n = 7), Icterohaemorrhagiae (n = 10), Pomona (n = 6), Sejroe (n = 7), and Tarassovi (n = 4). *Leptospira* serogroup panels which have been widely used for animal studies have included: Australis (n = 7), Ballum (n = 12), Canicola (n = 7), Grippotyphosa (n = 7), Hebdomadis (n = 8), Icterohaemorrhagiae (n = 14), Pomona (n = 12) and Sejroe (n = 9). The serogroups investigated for each animal group were not always detected as indicated in Fig 3 and S1 Table.

## Predominant *Leptospira* serogroups detected in human and animals

Thirty (n = 30) studies were able to report and describe serogroup diversity out of those studies using MAT, CAAT and molecular typing diagnostic approaches. The review found 17 *Leptospira* serogroups reported from humans and across animal species in Tanzania. In the case of humans, the most detected serogroups were Icterohaemorrhagiae (n = 11), Grippotyphosa (n = 8), Australis (n = 8), Ballum (n = 7), Hebdomadis (n = 6) and Sejroe (n = 6). We only counted the MAT serogroup once for samples that were used by multiple studies [19–21,27,30]. The most prevalent serogroups in people were Sejroe, Icterohaemorrhagiae and Australis Tables 1 and S2. The serogroups detected in the highest proportion of hospital patients were Australis, Icterhaemorrhagiae and Djasiman Tables 1 and S2.

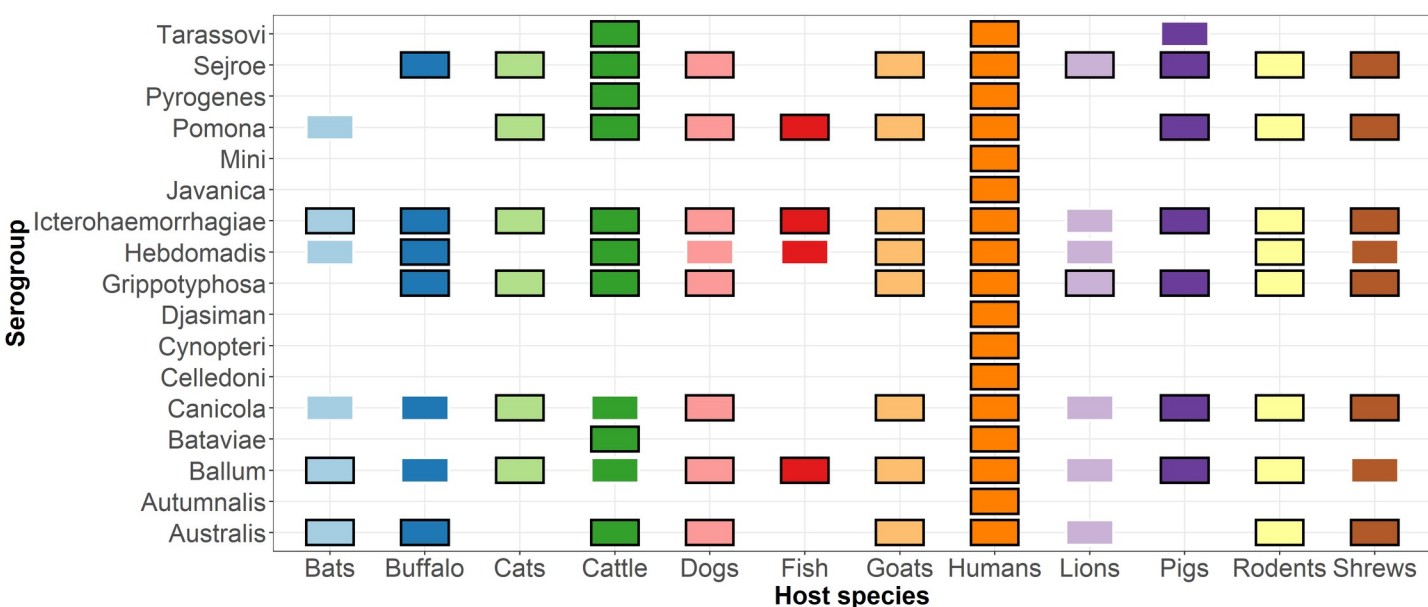

**Fig 3. Serogroups used in the Microscopic Agglutination Test (MAT) for detection of antibodies to *Leptospira* in humans and animals in Tanzania (1997–2019).** The coloured box indicates that samples were screened for these serogroups and the black outline indicates that the serogroup was detected (n = 28).

The most predominant *Leptospira* serogroups being reported in different animals were Icterohaemorrhagiae in 11 different animals (cattle, rodents, shrew, dogs, goat, sheep, bats, buffalo, pigs, cats and fish), Grippotyphosa and Sejroe in 10 animals (cattle, rodents, shrew, dogs, goat, sheep, buffalo, lion, cats, pigs), Pomona in 9 animals (cattle, rodents, shrew, dogs, goat, sheep, pigs, cats and fish), and Ballum in 8 animals (rodents, dogs, goats, sheep, bats, pigs, cats, and fish). The study carried out in wildlife found zero leptospiral antibodies in zebras which may be due to the small number of samples tested [11]. The most prevalent serogroups in rodents were Australis and Icterohaemorrhagiae and in cattle the most prevalent serogroups were Sejroe and Tarassovi Table 1.

## Leptospirosis and prevalence in humans

Leptospirosis in humans was reported by 20 eligible studies from 10 regions of Tanzania. Six of these studies were cross sectional studies investigating seroprevalence in the general population Table 2. The findings from two papers which conducted studies on people in Katavi from

**Table 1. Mean prevalence of antibodies to *Leptospira* serogroups in people in cross-sectional studies; in febrile patients; in rodents; in cattle in Tanzania in leptospirosis papers published 1997–2021.**

| Study type, number and references | Seroprevalence (%) | | | | | | | |
|---|---|---|---|---|---|---|---|---|
| | Australis | Ballum | Djasiman | Grippotyphosa | Hebdomadis | Icterohaemorrhagiae | Sejroe | Tarassovi |
| Cross-sectional studies in people (n = 5) [15,26,31–33] | 3.43 | 0.48 | NT | 1.58 | 1.50 | 5.38 | 9.35 | 1.00 |
| Hospital based studies in febrile patients (n = 4) [21,27,29,34] | 20.48 | 5.78 | 16.20 | 8.20 | 6.13 | 17.45 | 4.18 | 7.4 |
| Cross sectional studies in rodents (n = 7) [11,15,31,35–38] | 8.38 | 1.47 | NT | 2.07 | 0.28 | 7.29 | 0.37 | NT |
| Cross sectional studies in cattle (n = 6) [11,15,39–41] | 0.80 | 0.00 | NT | 4.80 | 5.10 | 4.25 | 15.94 | 15.10 |

NT—Not tested

**Table 2. Summary of studies reporting leptospirosis and seroprevalence of antibodies to *Leptospira* in humans in Tanzania 1997–2019.**

| Reference | Year | Study area | N | Seroprevalence (%) | Acute leptospirosis (%) | Risk factors/exposure |
|---|---|---|---|---|---|---|
| [15] | 1996 | Morogoro, Dar es Salaam, Mbeya, Kilimanjaro, Tanga, Singida and Mwanza | 375 | 0.3 | ND | ND |
| [33] | 2005 | Tanga | 199 | 15.1 | ND | ND |
| [11]* [26] | 2012–2013 | Katavi | 267 | 29.96 | ND | Slaughtering and handling of bush meat |
| [32] | 2017 | Mwanza | 250 | 10 | ND | Abattoir workers and meat vendors |
| [31] | No date | Kagera | 455 | 15.8 | ND | Fishing and working in sugarcane plantation |
| [18] | 1996–2006 | Morogoro | 506 | ND | 0.2 Patients | ND |
|  |  |  | 83 | ND | 3.6 Abattoir workers | ND |
| [30] | 2007–2008 | Kilimanjaro | 831 | ND | 8.4 | ND |
| [42] | 2008 | Dar es Salaam | 1005 | ND | 0.47 | ND |
| [34] | 2013 | Morogoro | 370 | ND | 11.6 | Heavy rain and presence of rodents in residential areas |
| [43] | 2014 | Morogoro | 191 | ND | 2 | ND |
| [23] | 2013–2014 | Dar es Salaam | 519 | ND | 0.2 | ND |
| [25] | 2014 | Morogoro | 842 | ND | 3 | ND |
| [21] | 2012–2014 | Kilimanjaro | 1293 | ND | 1.9 | Cleaning animal waste and rice farming |
| [29] | 2016–2017 | Arusha | 104 | ND | 5.8 | ND |

ND: not described

*Studies used the same data

2012–2014 are reported once [11,26]. From 1997–2019, a total of 209 out of 1546 tested samples were seropositive for antibodies against *Leptospira* spp. serogroups Table 2. The prevalence of antibodies to *Leptospira* varied depending on the study area, study design and interpretation of the results from 0.3% to 29.9%. Risk factors identified from 5 studies include occupational exposures such as contact with animals, animal waste and animal products Table 2.

There were also 14 hospital-based studies examining acute leptospirosis. Seven papers reported findings from the same patients in Kilimanjaro from 2007–2008 and/or 2013–2014 [10,19–21,27,28,30]. We have only reported the acute cases from these seven studies as defined by the authors and reported in 2 papers [21,27]. From 1997–2019 there were 173 acute cases of leptospirosis identified from 5661 febrile patients. Additionally, one study reported *Leptospira* in the urine of abattoir workers (3/83) which is not included in this number [18].

Leptospirosis incidence was estimated by two systematic hospital based and health care utilization surveys from the Kilimanjaro region. There was a large difference in the incidence estimations between the two studies. One study was conducted between 2007–2008 with the calculated incidence of acute leptospirosis ranging from 75–102 per 100,000 people annually [19]. The other study reported a lower leptospirosis incidence of 11–18 cases per 100,000 people annually from 2012–2014 [20].

## Animal leptospirosis and prevalence

Several leptospirosis studies have been carried out in various animal species in Tanzania, and in this review, a total of 18 studies met the inclusion criteria and were examined, 15 were cross sectional prevalence studies Table 3, 1 case control study [39] and 2 identified new serovars [44,45]. The total number of animals tested was 9090, though there were variations in the sample size and species between regions. The animals investigated were rodents (n = 10), shrews (n = 7), cattle (n = 8), goats (n = 3), pigs (n = 2), dogs (n = 2), bats (n = 2), sheep (n = 1), fish (n = 1), buffaloes (n = 1), lions (n = 1), zebra (n = 1). Eleven animal types were confirmed to have been exposed to *Leptospira*. These include rodents, shrews, cattle, goats, pigs, dogs, bats, sheep, fish, buffaloes, and lions Table 3. Among the animals studied, rodents (*Aesthomys chrysophilus*, *Dasmys incomtus*, *Mastomys natalensis*, *Rattus rattus*, *Lemniscomys griselda*, *Lemniscomys rosalia* and *Gerbilliscus vicinus*) were the most investigated followed by cattle in Tanzania. The prevalence of antibodies to *Leptospira* in cattle ranged from 5.6–51.0%, and in rodents from 1.8–25.8%. The presence of antibodies in serum samples was determined by MAT with recent studies adopting qPCR and molecular sequencing to confirm the infection from kidney samples for explorations of *Leptospira* serogroups diversity [10,22].

## Discussion

This review gives an insight on *Leptospira* prevalence and exposure, leptospirosis and the predominant *Leptospira* serogroups and their diversity in human and animal populations in Tanzania. It is evident after a detailed review of the published literature that leptospirosis is a prevalent zoonosis in Tanzania and present in various hosts including humans, livestock, wild animals, and aquatic life. There is an uneven distribution of research studies with large regions having inadequate or no leptospirosis information. The presence of universities or research institutions in regions that were overrepresented may reflect a degree of bias in the study site selection. For example, Kilimanjaro Clinical Research Institute (KCRI) conducted several human leptospirosis studies in the northern part of Tanzania while the Sokoine University of Agriculture conducted predominantly animal studies in the Morogoro region.

Our findings show that human leptospirosis is an important zoonosis of public health impact in Tanzania. Leptospirosis is widespread and prevalence varies between different settings and different populations. The actual burden of leptospirosis in humans may be difficult to estimate due to the limited and uneven distribution of studies and disease underestimation in the country. However, this trend is not unique to Tanzania, with the majority of low and middle-income countries (LMICs) facing similar challenges of inadequate surveillance data and diagnostic facilities [2]. Similar reports of leptospirosis prevalence as identified in this review have been reported in neighbouring countries. A study conducted in Kenya reported an apparent seropositivity of 13.4% in slaughterhouse workers [48], and a study of non-pregnant women in Uganda found 35% seropositive [49].

There was a large difference between the incidence reported in 2007–2008 and 2012–2014 in Kilimanjaro. This may be due to differences in the population selected, sample size or there may be variation in the leptospirosis incidence dependant on unknown factors [19,20]. Human leptospirosis in Tanzania may result from complex interactions between humans, animal carriers (such as cattle, rodents, dogs and pigs), and environments that favour perpetuation of leptospires and disease transmission.

The serological approaches utilized by various studies identified a diversity of *Leptospira* serogroups circulating in humans and animals. The MAT test was used in the majority of studies. MAT is a widely used diagnostic reference method for many studies, though not accessible in many laboratories due to its cost. MAT testing has many limitations: high levels of

**Table 3. Summary of studies reporting animals with leptospirosis in Tanzania 1997–2019.**

| Reference | Year | Study area | Animal species | N | Seroprevalence (%) | *Leptospira* detected by culture* or PCR** (%) |
|---|---|---|---|---|---|---|
| [15] | 1996 | Morogoro, Dar es salaam, Mbeya, Kilimanjaro, Tanga, Singida and Mwanza | Cattle | MAT n = 374 | 5.6 | |
| | | | Cattle | Culture n = 1021 | | 0.7* |
| | | | Dogs | 208 | 38 | |
| | | | Rodent | 537 | 1.8 | |
| [18] | 1996–2006 | Morogoro | Giant pouch rats | 285 | | 8.4* |
| | | | Field rats | 1382 | | 0.6* |
| | | | Shrews | 298 | | 3.7* |
| | | | Goats | 100 | 38 | |
| | | | Pigs | 100 | 41 | |
| | | | Dogs | 100 | 39 | |
| | | | Cats | 64 | 14.1 | |
| | | | Small rodents | 500 | 5 | |
| | | | Small rodents | 90 | 16.9 | |
| | | | African giant rats | 65 | 15.4 | |
| | | | Shrew | 4 | 25 | |
| [17] | 2003 | Morogoro | Fish | 48 | 54.2 | |
| [37] | No date | Morogoro | Rodent | 20 | 0 | 0* & 5** |
| | | | Shrew | 7 | 0 | 29* & 29** |
| [41] | 2002–2004 | Tanga | Cattle | 51 | 51 | |
| [40] | 2003–2004 | Tanga | Cattle | 655 | 30.3 | |
| [39] | 2005 | Tanga | Cattle | 80 | 21.3 | |
| [46] | 2007–2008 | Morogoro | Pig | MAT n = 385 | 4.4 | |
| | | | | Culture n = 236 | | 0.8* |
| [36] | 2007–2008 | Morogoro | Rodent and shrew | 348 | 17.8 | |
| [11] | 2012–2013 | Katavi | Cattle | 1103 | 30.37 | |
| | | | Goat | 248 | 8.47 | |
| | | | Rodent | 207 | 20.29 | |
| | | | Shrew | 11 | 9.09 | |
| | | | Buffalo | 38 | 28.95 | |
| | | | Lion | 2 | 50 | |
| | | | Zebra | 2 | 0 | |
| [16] | 2013 | Morogoro | Bat | 36 | 19.4 | |
| [38] | 2012–2013 | Morogoro | Rodent | 89 | 25.8 | |
| | | | Shrew | 1 | 100 | |
| [10] | 2013–2014 | Kilimanjaro | Cattle | 452 | | 7** |
| | | | Goat | 167 | | 1.2** |
| | | | Sheep | 89 | | 1.1** |
| | | | Rodents | 384 | | 0** |
| [47] | 2016–2017 | Morogoro | Dogs | 232 | 9.5 | |
| [35] | No date | Morogoro | Rodents | 70 | 22.9 | |

*(Continued)*

**Table 3.** (Continued)

| Reference | Year | Study area | Animal species | N | Seroprevalence (%) | *Leptospira* detected by culture* or PCR** (%) |
|---|---|---|---|---|---|---|
| [31] | No date | Kagera | Shrew and rodent | 24 | 16.7 | 0* |

detectable antibodies are needed for a positive result and usually do not occur before the fourth week after disease onset [50] and it is time consuming and labour intensive [51,52]. Despite these drawbacks, the MAT test remains the only gold standard serological test and is considered a reference diagnostic test for leptospirosis in many settings [6,53].

The review found a large variation in the serogroup panels and the definition of positivity used across the studies S1 Table. When establishing a diagnostic panel it is advisable to include locally circulating serogroups or if these are not known to include a wide panel of pathogenic serogroups [6]. A list of candidate *Leptospira* serovars for diagnosis of leptospirosis using MAT in the African region was recently published based on research conducted in Tanzania [18]. However, emergence of new serovars suggests widening the serovar panels [14].

Most serogroups detected in animal species in the reviewed studies were also reported in humans. The most prevalent serogroups detected in rodents were Australis and Icterohaemoraghiae and Sejroe in cattle. These were also the most prevalent serogroups detected in people. This suggests that rodents and cattle may be an important source of infection in these settings. However, it is difficult to demonstrate transmission between animals and humans in our review because of the variability in the serogroup panel and different study designs.

A variety of domestic and wild animals in eighteen studies provide evidence of leptospirosis infections in animal populations in Tanzania. The review suggests that the main animal reservoirs for human leptospirosis may vary across the country, with primarily cattle, rodents, pigs, and dogs playing significant roles in disease transmission to humans. Rodents are important reservoirs of pathogenic *Leptospira* in many settings [12]. This review identified 10 studies reporting evidence of *Leptospira* in rodents and a diverse range of serogroups were detected Fig 3. There were only two studies in which *Leptospira* was detected in the sampled rodents using culture and PCR [18,37]. The lack of evidence of *Leptospira* in rodents in other studies using culture and qPCR techniques may indicate a methodological problem or lack of infected animals [22]. This scenario has also been reported in other studies, though such studies were associated with a limited sample size [12]. There may be differences in the prevalence of *Leptospira* in rodents between regions and between rural and urban settings [54]. Inappropriate sampling technique, sample preservation and an inadequate number of micro-organisms or loss of bacteria during culture can lead to false negative results.

Among exposed animals, cattle had the highest seropositivity, though this varied depending on geographical area. Cattle may be potential reservoirs and sources of human infection in Tanzania, particularly in rural areas where the majority of residents are smallholder dairy farmers and pastoralists [55]. Cattle are an important maintenance host for serogroup Sejroe [1,56] and transmission to farm workers and slaughtermen has been documented [48,57]. Animal contact particularly occupational exposures was identified as a risk factor by the reviewed papers and this is likely to have an important role in the epidemiology of leptospirosis in people in Tanzania [11,21,31,32].

## Conclusion and recommendation

This review provides a summary of important information on the prevalence and distribution of the predominant *Leptospira* serogroups in humans and animals in Tanzania. Our review

suggests that more comprehensive leptospirosis studies are needed in rodents and livestock across different agro-ecological zones for a deeper understanding of the epidemiology and to understand the risks of human leptospirosis for better management and control of the disease. The role of livestock in disease transmission among the smallholder farmers and other risk factors for human leptospirosis should be well studied for future disease control plans.

In most studies conducted in Tanzania, the MAT is the only diagnostic test used widely for leptospirosis detection however MAT may be impractical in many clinical laboratories due to the cost and complexity [52]. An alternative tool, such as rapid diagnostic tests (RDTs), was proposed by a recent policy brief and may be appropriate in a clinical setting for routine screening of patients with non-malaria fever [58]. The performance of RDTs is variable and would need to be trialled before implementation [28,59]. Raising awareness among health providers and the community on leptospirosis is recommended as a vital strategy for disease control and prevention.

## Supporting information

**S1 Table. Summary of the papers included in this review of leptospirosis in Tanzania 1997–2019 including year of research, study design, geographical location, target populations, diagnostics tests, and results for each study.**
(DOCX)

**S2 Table.** A) Prevalence of antibodies to *Leptospira* serogroups in people in cross-sectional studies; B) in febrile patients; C) in rodents; D) in cattle in Tanzania in leptospirosis papers published 1997–2021.
(DOCX)

## Acknowledgments

We thankfully recognize the Nelson Mandela African Institution of Science and Technology (NM-AIST) for providing the necessary infrastructure to complete this review study. The authors also would like to thank the International Livestock Research Institute (ILRI) for organizing training and guidance on database and literature search strategies of this study. Special thanks to researchers from the International Livestock Research Institute, The Roslin Institute, and the SRUC for their support.

The findings and conclusions contained within are those of the authors and do not necessarily reflect positions or policies of the Bill & Melinda Gates Foundation nor the UK Government.

## Author Contributions

**Conceptualization:** Shabani Kiyabo Motto.

**Data curation:** Shabani Kiyabo Motto, Elizabeth Anne Jessie Cook.

**Formal analysis:** Shabani Kiyabo Motto, Elizabeth Anne Jessie Cook.

**Methodology:** Shabani Kiyabo Motto.

**Supervision:** Gabriel Mkilema Shirima, Barend Mark de Clare Bronsvoort, Elizabeth Anne Jessie Cook.

**Writing – original draft:** Shabani Kiyabo Motto.

**Writing – review & editing:** Gabriel Mkilema Shirima, Barend Mark de Clare Bronsvoort, Elizabeth Anne Jessie Cook.

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
