## [Decision Letter · Decision Letter 0]

7 Mar 2021

Dear Mr Motto,

Thank you very much for submitting your manuscript "Epidemiology of leptospirosis in Tanzania: A review of the current status, serovar diversity and reservoirs" for consideration at PLOS Neglected Tropical Diseases. As with all papers reviewed by the journal, your manuscript was reviewed by members of the editorial board and by several independent reviewers. In light of the reviews (below this email), we would like to invite the resubmission of a significantly-revised version that takes into account the reviewers' comments. 

Reviewer #1: Thank you for the opportunity to review your systematic review. It was exciting to see a summary of recent research and a pleasure to read. I have some suggestions about areas in which the manuscript could be improved.

MAJOR:

1. Search strategy - You seem to be missing a number of relevant studies. It is not clear from the published search strategy why this is the case. When I use your search terms in Google I find additional references and understanding how/why these were excluded is important. For example:

a. Allan KJ, Maze MJ, Galloway RL, Rubach MP, Biggs HM, Halliday JE, Cleaveland S, Saganda W, Lwezaula BF, Kazwala RR, Mmbaga BT. Molecular Detection and Typing of Pathogenic Leptospira in Febrile Patients and Phylogenetic Comparison with Leptospira Detected among Animals in Tanzania. The American Journal of Tropical Medicine and Hygiene. 2020 Oct;103(4):1427. - contains additional molecular data from Kilimanjaro Region

b. Maze MJ. The Impact of Leptospirosis in Northern Tanzania: A Thesis Submitted for the Degree of Doctor of Philosophy at the University of Otago, Dunedin, New Zealand (Doctoral dissertation, University of Otago). This thesis contains a chapter on leptospirosis in the Ngorongoro Conservation Area.

c. D'acremont V, Kilowoko M, Kyungu E, Philipina S, Sangu W, Kahama-Maro J, Lengeler C, Cherpillod P, Kaiser L, Genton B. Beyond malaria—causes of fever in outpatient Tanzanian children. New England Journal of Medicine. 2014 Feb 27;370(9):809-17. This study investigated leptospirosis among children with fever

d. Boillat-Blanco N, Mbarack Z, Samaka J, Mlaganile T, Kazimoto T, Mamin A, Genton B, Kaiser L, D'Acremont V. Causes of fever in Tanzanian adults attending outpatient clinics: a prospective cohort study. Clinical Microbiology and Infection. 2020 Sep 4. This study used PCR to identify leptospirosis.

2. Leptospira exposure (ie seropositivity) vs leptospirosis (ie the acute disease) - It is important to clearly differentiate between the two as they are very different things. The lack of clarity is evident in line 299 - the Biggs 2011 study found 8.8% of patients hospitalized with a fever had acute leptospirosis with an additional 33.3% (ie 42.1% in total) seropositive. I suggest that you need to review all your papers regarding human infection again and consider how they should be interpreted. The interpretation is of course complex because acute leptospirosis would be unexpected in a cross sectional survey, and the incidence can only really be assessed by investigating those with a compatible illness.

3. Differentiating absent from not investigated. When considering MAT results I think you need to distinguish between serogroups that are not identified because they were not included in a panel and those that were included but tested negative. The entire section around predominant serovars needs reconsideration. In addition, Figure 5 is (I think) potentially misleading. As an alternative I think you should consider whether to report the prevalence of seropositivity by serogroup.

MINOR

General - Please review the written English within the paper - for example 'biasness' (line163) and numerous examples of disagreement between pronoun and noun (singular vs plural) and missing prepositions. Apologies for bringing this up, but the errors are to the point that they affect reading.

INTRODUCTION:

1. There are some sentences (eg lines 82-83 about leptospirosis in Malaysia) and references (eg reference 5 the case report from Russia) that seemed of limited relevance to the topic. I suggest focusing comments and references towards Tanzania.

2. Lines 82-83 regarding modes of transmission fails to mention contact with a contaminated environment as a potential source of infection.

Methods:

1. Including the search date is important (see above).

2. The ILS meeting is probably the most important scientific meeting for identifying leptospirosis abstracts and as far as I am aware it does not put the submitted abstracts on PubMed or Google. Did you have a strategy for reviewing these?

Results:

In addition to the 2 major comments above, 

1. Lines 246-247. The statement regarding the confirmed leptospirosis cases needs clarification. What is the definition? Where has the denominator come from? As above, a cross sectional seroprevalence study should not be expected to identify acute leptospirosis cases and shouldn't feature as part of a denominator.

2. Lines 256-263 - When considering the variation in estimated leptospirosis incidence you highlight study design, population sampling and study size, but neglect to consider variation in leptospirosis incidence. Is this not possible?

Discussion:

1. Line 291 - While I am sympathetic to the perspective that leptospirosis doesn't get enough attention, I think the finding does need some support from your data. You have identified 30 studies - most published within the last decade. How much attention does it deserve, and from whom?

2. Lines 294-296 - The discussion about sources of leptospirosis does not mention rodents. Several authors have identified seropositive rodents and Mgode et al have isolated Leptospira from rodents. Totally discounting them seems a little premature.

3. Incidence of human leptospirosis - perhaps an alternative explanation of the data is that the incidence of leptospirosis varies by location and over time. The assertion that it has increased (line 298) is not supported by the only 2 actual datapoints on acute leptospirosis incidence (Biggs 2013 and Maze 2016).

4. Lines 324-327 - These seem to say that there has been only one study of rodents and it was negative. Your review shows this to be false. Mgode et al have identified Leptospira in rodents. Further you cite a number of technical errors that may have accounted for it. What about the possibility that Leptospira weren't (often) present in rats in Kilimanjaro region at the time of the study? It is notable that Allan et al did not trap many (if any) Mastomys species, whereas Mgode cultured Leptospira from them. There is precedent (a paper from Thailand last year I think) for low prevalence of leptospira carriage among peri-urban Rattus species, but high prevalence among rural rice field rats (Losea species from memory).

5. The paragraph on serologic approaches could more concisely state that MAT does not distinguish between infecting serovars. I think this is what you are getting at, but it is not clear.

6. The conclusion around RDTs is appealing, but the challenge is that (in my opinion at any rate) there isn't a good available RDT. While I appreciate that there isn't a lot of published data on this yet (there was an abstract at the ILS in 2017 reporting poor performance of lepto IgM RDTs in Kilimanjaro Region) your data doesn't really support this statement.

REFERNCES

There are numerous errors (eg non italisized genus names) and a mish-mash of referencing styles and text (including all caps).

Reviewer #2: This is an interesting review on leptospirosis in Tanzania that could also be representing the situation in African region where this disease is highly neglected.

The search seem to have skewed to mainly open access journals that full articles were retrievable and some important hosts have not been fully considered.

Relevants comments can be found on respective sections and lines. Generally the work is worth and increases awareness and knowledge on this disease in the region.

ABSTRACT

Line 53: ...typhoid and brucellosis. should be typhoid, brucellosis and other diseases. Delete Despite in the next sentence in this line to start with "More than 250 pathogenic.....

Line 56: leptospira should be Leptospira 

INTRODUCTION

Line 73: ....including Tanzania (10) where the disease.... This citation number 10 refers to Thailand not Tanzania.

Line 79: ......"incidences of 72-102 and 529 leptospiral cases....." a denominator for the 529 might be useful here.

Line 82-83: .The high leptospiral cases in Malaysia.......296 deaths in 5 years: this sentence is not clear what it refers needs rephrasing and linking it well with the rest of contents in this paragraph.

Line 87: .....non-pathogenic Leptospira (25, 26): Original references would fit better here, or add to these ones.

METHODOLOGY - 

RESULTS:

Line 178: ....in humans (n=8) needs references. similarly, ...........two studies on animals (cattle, goat and rodents) - needs references as well.e.g. the two studies could be cited here.

Line 226: ......serovars serotyped: should be serovars reported.

Line 264: table 1: the locality named Moshi in column 3 should preferably be named as Moshi Kilimanjaro because the rest of the areas listed in this Table represent the regions while Moshi is an area in Kilimanjaro region.

Line 272: see comment on Moshi. should be Moshi Kilimanjaro.

Line 277: .....Lemniscomys Griselda should be Lemniscomys griselda. the species name is small letter.

Line 282: Table 2. omit DFM among the listed test - it is not a test but a tool for examining leptospires 

Row number 19 column no.5 the 455 are humans not rodents and shrews. In this study (39) there were rodents and shrews but the 455 are humans.

For citation no. 59 (row no. 14 column 4 add MAT and culture. 

This Table can be enriched by including information from a study by Mgode et al (2015) published in PLOSNTD which gives insights on leptospirosis situation in broad range of reservoir hosts including data on cattle, cats, goats, sheep, pigs, rodents and shrews. Another citation that reports genetic diversity of Leptospira isolates from Tanzania published by Ahmed et al (2006) could also add important information on Leptospira serovars reported from Tanzania which were described/identified using multilocus sequence typing. This paper is missing in this manuscript and has a large/ in-depth report of Leptospira isolates from Tanzania (DOI: 10.1186/1476-0711-5-28).

DISCUSSION:

Line 290.... it has not received sufficient attention.....This probably requires rephrasing because some work on leptospirosis in Tanzania has been used/cited in the National One Health strategic plan in Tanzania and East Africa community (specifically the work on leptospirosis in bats from Morogoro Tanzania, as well as publication on Leptospira serovar Sokoine from cattle. However, much would have been done given the amount of evidence that is available on leptospirosis and Leptospira from Tanzania. A policy brief on Leptospirosis in Tanzania is also in place (on researchgate) which highlights the situation of this disease in humans in Tanzania - probably contributes to the increasing awareness of non-malarial fevers. 

Line 297: For the past 9, ... this is incomplete, 9 what? years?

Line 306: ...establish = established

Line 368: ..... a list of candidate Leptospira serovars for diagnosis of leptospirosis using MAT in African region has been published (Mgode et al. 2015) which is based on data obtained from use of local serovars vs imported serovars. Similarly, policy brief (2016) recommending introduction of rapid diagnostic kits for leptospirosis has been recommended for patients with non-malarial fevers (DOI: 10.13140/RG.2.2.13464.60165) could be relevant in this discussion line 373.

REFERENCES: 

Line 460: citation no. 23: seem to be incomplete

Reviewer #3: This manuscript synthesises what is known about the epidemiology of leptospirosis through a literature review. There is some confusion throughout the manuscript in the use of prevalence (presence of pathogen) and seroprevalence (presence of antibodies to the pathogen). The subject area is important and relevant and justifies publication.

Line 14-15: "Leptospirosis is…" - Awkward sentence. Try rewording 

Line 34-36: Previously stated that 8 studies were undertaken in Kiliminjaro but this adds to 10

Line 37: "Sejroe" not "Serjoe"

Line 41-43: State the range in prevalence to provide some detail on how large the variation is

Line 77: "subacute and long-lasting illness" - I'm not sure that these are the most appropriate references they just reference other studies which state this. Cite the initial studies where this was shown

Line 81-83: Not sure why this is relevant

Line 84: It is more than 250 serovars (as stated in abstract)

Line 86: It is many more than 21 species. See recent paper https://doi.org/10.1371/journal. pntd.0007270 

line 169-178: When describing the number of studies focussing on each host type in each region these do not add up to the number of studies performed in each region (line 169-170)

line 181: "serological diagnostic techniques" - many of these examples (culture, PCR) are not serological techniques

line 184 -186: This doesn't make sense. All the tests are diagnostic tests and I'm not sure what "widely recognised" means

line 180-203: I don't think this figure is publication quality yet and I think it would be worth that providing references for each of the assays (perhaps as a supplementary table)

line 209: If some serovars were frequently used then please list which ones

line 218-219: provide a reference

line 219-221: This is for discussion not results

Figure 4: I couldn't make sense of this figure. It needs to be improved

Line 230-232: Confusing sentence. Not sure what the point is. Is it that lepto has been detected in every species except zebra or that it hasn't been detected in zebra because few animals were sampled?

Line 246: I think this is misleading. It give the impression that all 7129 cases were suspected leptospirosis and only 576 were confirmed

Line 262-263: The most likely possibility is that lepto incidence truly did decrease

Line 279: presence of leptospires is not detected by MAT. Only leptospire antibodies are detected by MAT

Line 297-298: "For the past 9.." something has been left out of this sentence

Line 306: "leptospirosis" not "Leptospirosis"

Line 306: "base level" - I am not clear that a base level prevalence is established in this study? To what does this refer?

Line 313: To what does n=16 refer?

Line 315: dogs aren't livestock

Line 327: Again. The most likely cause of this finding is that the rodents weren't infected. It isn't uncommon as you point out. There doesn't have to be a methodology problem.

Line 338-339: Risk of contamination of what? I'm not sure the author fully understands MAT

Line 339-340: "MAT cannot be considered as a gold standard test" - but it is considered the gold standard serological test unless the author wishes to propose another test as gold standard

Line 349-351: I think the discussion should include human exposure identified in clinical cases (fever) and human cases identified in surveillance (no fever). There are some serovars (Hardjo) that might not cause severe human disease.

We cannot make any decision about publication until we have seen the revised manuscript and your response to the reviewers' comments. Your revised manuscript is also likely to be sent to reviewers for further evaluation.

Sincerely,

Alan J A McBride, Ph.D.

Associate Editor

Melissa Caimano

Deputy Editor

Reviewer #1: Thank you for the opportunity to review your systematic review. It was exciting to see a summary of recent research and a pleasure to read. I have some suggestions about areas in which the manuscript could be improved.

MAJOR:

1. Search strategy - You seem to be missing a number of relevant studies. It is not clear from the published search strategy why this is the case. When I use your search terms in Google I find additional references and understanding how/why these were excluded is important. For example:

a. Allan KJ, Maze MJ, Galloway RL, Rubach MP, Biggs HM, Halliday JE, Cleaveland S, Saganda W, Lwezaula BF, Kazwala RR, Mmbaga BT. Molecular Detection and Typing of Pathogenic Leptospira in Febrile Patients and Phylogenetic Comparison with Leptospira Detected among Animals in Tanzania. The American Journal of Tropical Medicine and Hygiene. 2020 Oct;103(4):1427. - contains additional molecular data from Kilimanjaro Region

b. Maze MJ. The Impact of Leptospirosis in Northern Tanzania: A Thesis Submitted for the Degree of Doctor of Philosophy at the University of Otago, Dunedin, New Zealand (Doctoral dissertation, University of Otago). This thesis contains a chapter on leptospirosis in the Ngorongoro Conservation Area.

c. D'acremont V, Kilowoko M, Kyungu E, Philipina S, Sangu W, Kahama-Maro J, Lengeler C, Cherpillod P, Kaiser L, Genton B. Beyond malaria—causes of fever in outpatient Tanzanian children. New England Journal of Medicine. 2014 Feb 27;370(9):809-17. This study investigated leptospirosis among children with fever

d. Boillat-Blanco N, Mbarack Z, Samaka J, Mlaganile T, Kazimoto T, Mamin A, Genton B, Kaiser L, D'Acremont V. Causes of fever in Tanzanian adults attending outpatient clinics: a prospective cohort study. Clinical Microbiology and Infection. 2020 Sep 4. This study used PCR to identify leptospirosis.

2. Leptospira exposure (ie seropositivity) vs leptospirosis (ie the acute disease) - It is important to clearly differentiate between the two as they are very different things. The lack of clarity is evident in line 299 - the Biggs 2011 study found 8.8% of patients hospitalized with a fever had acute leptospirosis with an additional 33.3% (ie 42.1% in total) seropositive. I suggest that you need to review all your papers regarding human infection again and consider how they should be interpreted. The interpretation is of course complex because acute leptospirosis would be unexpected in a cross sectional survey, and the incidence can only really be assessed by investigating those with a compatible illness.

3. Differentiating absent from not investigated. When considering MAT results I think you need to distinguish between serogroups that are not identified because they were not included in a panel and those that were included but tested negative. The entire section around predominant serovars needs reconsideration. In addition, Figure 5 is (I think) potentially misleading. As an alternative I think you should consider whether to report the prevalence of seropositivity by serogroup.

MINOR

General - Please review the written English within the paper - for example 'biasness' (line163) and numerous examples of disagreement between pronoun and noun (singular vs plural) and missing prepositions. Apologies for bringing this up, but the errors are to the point that they affect reading.

INTRODUCTION:

1. There are some sentences (eg lines 82-83 about leptospirosis in Malaysia) and references (eg reference 5 the case report from Russia) that seemed of limited relevance to the topic. I suggest focusing comments and references towards Tanzania.

2. Lines 82-83 regarding modes of transmission fails to mention contact with a contaminated environment as a potential source of infection.

Methods:

1. Including the search date is important (see above).

2. The ILS meeting is probably the most important scientific meeting for identifying leptospirosis abstracts and as far as I am aware it does not put the submitted abstracts on PubMed or Google. Did you have a strategy for reviewing these?

Results:

In addition to the 2 major comments above, 

1. Lines 246-247. The statement regarding the confirmed leptospirosis cases needs clarification. What is the definition? Where has the denominator come from? As above, a cross sectional seroprevalence study should not be expected to identify acute leptospirosis cases and shouldn't feature as part of a denominator.

2. Lines 256-263 - When considering the variation in estimated leptospirosis incidence you highlight study design, population sampling and study size, but neglect to consider variation in leptospirosis incidence. Is this not possible?

Discussion:

1. Line 291 - While I am sympathetic to the perspective that leptospirosis doesn't get enough attention, I think the finding does need some support from your data. You have identified 30 studies - most published within the last decade. How much attention does it deserve, and from whom?

2. Lines 294-296 - The discussion about sources of leptospirosis does not mention rodents. Several authors have identified seropositive rodents and Mgode et al have isolated Leptospira from rodents. Totally discounting them seems a little premature.

3. Incidence of human leptospirosis - perhaps an alternative explanation of the data is that the incidence of leptospirosis varies by location and over time. The assertion that it has increased (line 298) is not supported by the only 2 actual datapoints on acute leptospirosis incidence (Biggs 2013 and Maze 2016).

4. Lines 324-327 - These seem to say that there has been only one study of rodents and it was negative. Your review shows this to be false. Mgode et al have identified Leptospira in rodents. Further you cite a number of technical errors that may have accounted for it. What about the possibility that Leptospira weren't (often) present in rats in Kilimanjaro region at the time of the study? It is notable that Allan et al did not trap many (if any) Mastomys species, whereas Mgode cultured Leptospira from them. There is precedent (a paper from Thailand last year I think) for low prevalence of leptospira carriage among peri-urban Rattus species, but high prevalence among rural rice field rats (Losea species from memory).

5. The paragraph on serologic approaches could more concisely state that MAT does not distinguish between infecting serovars. I think this is what you are getting at, but it is not clear.

6. The conclusion around RDTs is appealing, but the challenge is that (in my opinion at any rate) there isn't a good available RDT. While I appreciate that there isn't a lot of published data on this yet (there was an abstract at the ILS in 2017 reporting poor performance of lepto IgM RDTs in Kilimanjaro Region) your data doesn't really support this statement.

REFERNCES

There are numerous errors (eg non italisized genus names) and a mish-mash of referencing styles and text (including all caps).

Reviewer #2: This is an interesting review on leptospirosis in Tanzania that could also be representing the situation in African region where this disease is highly neglected.

The search seem to have skewed to mainly open access journals that full articles were retrievable and some important hosts have not been fully considered.

Relevants comments can be found on respective sections and lines. Generally the work is worth and increases awareness and knowledge on this disease in the region.

ABSTRACT

Line 53: ...typhoid and brucellosis. should be typhoid, brucellosis and other diseases. Delete Despite in the next sentence in this line to start with "More than 250 pathogenic.....

Line 56: leptospira should be Leptospira 

INTRODUCTION

Line 73: ....including Tanzania (10) where the disease.... This citation number 10 refers to Thailand not Tanzania.

Line 79: ......"incidences of 72-102 and 529 leptospiral cases....." a denominator for the 529 might be useful here.

Line 82-83: .The high leptospiral cases in Malaysia.......296 deaths in 5 years: this sentence is not clear what it refers needs rephrasing and linking it well with the rest of contents in this paragraph.

Line 87: .....non-pathogenic Leptospira (25, 26): Original references would fit better here, or add to these ones.

METHODOLOGY - 

RESULTS:

Line 178: ....in humans (n=8) needs references. similarly, ...........two studies on animals (cattle, goat and rodents) - needs references as well.e.g. the two studies could be cited here.

Line 226: ......serovars serotyped: should be serovars reported.

Line 264: table 1: the locality named Moshi in column 3 should preferably be named as Moshi Kilimanjaro because the rest of the areas listed in this Table represent the regions while Moshi is an area in Kilimanjaro region.

Line 272: see comment on Moshi. should be Moshi Kilimanjaro.

Line 277: .....Lemniscomys Griselda should be Lemniscomys griselda. the species name is small letter.

Line 282: Table 2. omit DFM among the listed test - it is not a test but a tool for examining leptospires 

Row number 19 column no.5 the 455 are humans not rodents and shrews. In this study (39) there were rodents and shrews but the 455 are humans.

For citation no. 59 (row no. 14 column 4 add MAT and culture. 

This Table can be enriched by including information from a study by Mgode et al (2015) published in PLOSNTD which gives insights on leptospirosis situation in broad range of reservoir hosts including data on cattle, cats, goats, sheep, pigs, rodents and shrews. Another citation that reports genetic diversity of Leptospira isolates from Tanzania published by Ahmed et al (2006) could also add important information on Leptospira serovars reported from Tanzania which were described/identified using multilocus sequence typing. This paper is missing in this manuscript and has a large/ in-depth report of Leptospira isolates from Tanzania (DOI: 10.1186/1476-0711-5-28).

DISCUSSION:

Line 290.... it has not received sufficient attention.....This probably requires rephrasing because some work on leptospirosis in Tanzania has been used/cited in the National One Health strategic plan in Tanzania and East Africa community (specifically the work on leptospirosis in bats from Morogoro Tanzania, as well as publication on Leptospira serovar Sokoine from cattle. However, much would have been done given the amount of evidence that is available on leptospirosis and Leptospira from Tanzania. A policy brief on Leptospirosis in Tanzania is also in place (on researchgate) which highlights the situation of this disease in humans in Tanzania - probably contributes to the increasing awareness of non-malarial fevers. 

Line 297: For the past 9, ... this is incomplete, 9 what? years?

Line 306: ...establish = established

Line 368: ..... a list of candidate Leptospira serovars for diagnosis of leptospirosis using MAT in African region has been published (Mgode et al. 2015) which is based on data obtained from use of local serovars vs imported serovars. Similarly, policy brief (2016) recommending introduction of rapid diagnostic kits for leptospirosis has been recommended for patients with non-malarial fevers (DOI: 10.13140/RG.2.2.13464.60165) could be relevant in this discussion line 373.

REFERENCES: 

Line 460: citation no. 23: seem to be incomplete

Reviewer #3: This manuscript synthesises what is known about the epidemiology of leptospirosis through a literature review. There is some confusion throughout the manuscript in the use of prevalence (presence of pathogen) and seroprevalence (presence of antibodies to the pathogen). The subject area is important and relevant and justifies publication.

Line 14-15: "Leptospirosis is…" - Awkward sentence. Try rewording 

Line 34-36: Previously stated that 8 studies were undertaken in Kiliminjaro but this adds to 10

Line 37: "Sejroe" not "Serjoe"

Line 41-43: State the range in prevalence to provide some detail on how large the variation is

Line 77: "subacute and long-lasting illness" - I'm not sure that these are the most appropriate references they just reference other studies which state this. Cite the initial studies where this was shown

Line 81-83: Not sure why this is relevant

Line 84: It is more than 250 serovars (as stated in abstract)

Line 86: It is many more than 21 species. See recent paper https://doi.org/10.1371/journal. pntd.0007270 

line 169-178: When describing the number of studies focussing on each host type in each region these do not add up to the number of studies performed in each region (line 169-170)

line 181: "serological diagnostic techniques" - many of these examples (culture, PCR) are not serological techniques

line 184 -186: This doesn't make sense. All the tests are diagnostic tests and I'm not sure what "widely recognised" means

line 180-203: I don't think this figure is publication quality yet and I think it would be worth that providing references for each of the assays (perhaps as a supplementary table)

line 209: If some serovars were frequently used then please list which ones

line 218-219: provide a reference

line 219-221: This is for discussion not results

Figure 4: I couldn't make sense of this figure. It needs to be improved

Line 230-232: Confusing sentence. Not sure what the point is. Is it that lepto has been detected in every species except zebra or that it hasn't been detected in zebra because few animals were sampled?

Line 246: I think this is misleading. It give the impression that all 7129 cases were suspected leptospirosis and only 576 were confirmed

Line 262-263: The most likely possibility is that lepto incidence truly did decrease

Line 279: presence of leptospires is not detected by MAT. Only leptospire antibodies are detected by MAT

Line 297-298: "For the past 9.." something has been left out of this sentence

Line 306: "leptospirosis" not "Leptospirosis"

Line 306: "base level" - I am not clear that a base level prevalence is established in this study? To what does this refer?

Line 313: To what does n=16 refer?

Line 315: dogs aren't livestock

Line 327: Again. The most likely cause of this finding is that the rodents weren't infected. It isn't uncommon as you point out. There doesn't have to be a methodology problem.

Line 338-339: Risk of contamination of what? I'm not sure the author fully understands MAT

Line 339-340: "MAT cannot be considered as a gold standard test" - but it is considered the gold standard serological test unless the author wishes to propose another test as gold standard

Line 349-351: I think the discussion should include human exposure identified in clinical cases (fever) and human cases identified in surveillance (no fever). There are some serovars (Hardjo) that might not cause severe human disease.
---

## [Decision Letter · Decision Letter 1]

19 Aug 2021

Dear Mr Motto,

Thank you very much for submitting your manuscript "Epidemiology of leptospirosis in Tanzania: A review of the current status, serogroup diversity and reservoirs" for consideration at PLOS Neglected Tropical Diseases. As with all papers reviewed by the journal, your manuscript was reviewed by members of the editorial board and by several independent reviewers. The reviewers appreciated the attention to an important topic. Based on the reviews, we are likely to accept this manuscript for publication, providing that you modify the manuscript according to the review recommendations. 

Reviewer #1: Thank you for the opportunity to review this revised manuscript. My concerns from the initial paper have for the most part been addressed. Having read the revised manuscript, I have a small number of additional questions/ comments.

Introduction:

Line 74: Including cattle, rodents and pigs - all true. Is there a role for dogs, sheep or goats?

Methods: Search strategy - I worry that the PubMed search strategy is too narrow. That it only identified 47 articles compared to 3720 identified in Google attests to that. It is perhaps not feasible to re-engineer your searches, but perhaps acknowledging the discrepancy between identified articles in Google vs PubMed and the potential to have missed articles that were not identified by the Google search would be appropriate. 

Discussion - The claim of primacy isn't entirely true. What about Allan et al and De Vries et al who both conducted systematic reviews of the entire continent. Your review provides greater detail for Tanzania - but isn't the first to address the topic.

Reviewer #3: Dear authors

Thank you for your revisions. Please see below a few suggestions.

Line 81-83: Check this. There are 17 species in P1 and 21 in P2

Line 87: "small prevalence" should be "low prevalence"

Lines 178-184: These sentences are more discussion than results

Lines 208-209: Molecular typing is not a diagnostic technique. It is way of identifying genetic diversity

Line 215: Should be Figure 3

Figure 3: "Javonica" should be "Javanica"

Line 227: "serovars" should be "serogroups"

Line 278: Please be careful using incidence rather than prevalence. Incidence always includes a time period e.g. XX cases/ 100000 individuals/year. I don't think any of these animal studies describe incidence.

Lines313-314: Please include the references

Supplementary Table 1: 

This table needs to be checked. I am unclear what the serogroup (serovar) column indicates. I have checked a few references and for reference 2 (Allan et al 2020) the serogroups refer to those identified as causing human infections whereas reference 4 (Biggs et al 2011) appears to show all the serogroups included in the MAT. Only 8 of these serogroups were identified in human cases. I notice that in later references it is shown whether serogroups were not detected. Perhaps this was an oversight for earlier references?

Under the PCR column some studies list DNA fingerprinting - this isn't PCR. There is also reference to TAC. I'm not sure what this is? Perhaps it should be defined in the footnotes? In addition, for some references e.g. 22 the results from hosts are shown in the PCR column.

Sincerely,

Alan J A McBride, Ph.D.

Associate Editor

Melissa Caimano

Deputy Editor

Reviewer #1: Thank you for the opportunity to review this revised manuscript. My concerns from the initial paper have for the most part been addressed. Having read the revised manuscript, I have a small number of additional questions/ comments.

Introduction:

Line 74: Including cattle, rodents and pigs - all true. Is there a role for dogs, sheep or goats?

Methods: Search strategy - I worry that the PubMed search strategy is too narrow. That it only identified 47 articles compared to 3720 identified in Google attests to that. It is perhaps not feasible to re-engineer your searches, but perhaps acknowledging the discrepancy between identified articles in Google vs PubMed and the potential to have missed articles that were not identified by the Google search would be appropriate. 

Discussion - The claim of primacy isn't entirely true. What about Allan et al and De Vries et al who both conducted systematic reviews of the entire continent. Your review provides greater detail for Tanzania - but isn't the first to address the topic.

Reviewer #3: Dear authors

Thank you for your revisions. Please see below a few suggestions.

Line 81-83: Check this. There are 17 species in P1 and 21 in P2

Line 87: "small prevalence" should be "low prevalence"

Lines 178-184: These sentences are more discussion than results

Lines 208-209: Molecular typing is not a diagnostic technique. It is way of identifying genetic diversity

Line 215: Should be Figure 3

Figure 3: "Javonica" should be "Javanica"

Line 227: "serovars" should be "serogroups"

Line 278: Please be careful using incidence rather than prevalence. Incidence always includes a time period e.g. XX cases/ 100000 individuals/year. I don't think any of these animal studies describe incidence.

Lines313-314: Please include the references

Supplementary Table 1: 

This table needs to be checked. I am unclear what the serogroup (serovar) column indicates. I have checked a few references and for reference 2 (Allan et al 2020) the serogroups refer to those identified as causing human infections whereas reference 4 (Biggs et al 2011) appears to show all the serogroups included in the MAT. Only 8 of these serogroups were identified in human cases. I notice that in later references it is shown whether serogroups were not detected. Perhaps this was an oversight for earlier references?

Under the PCR column some studies list DNA fingerprinting - this isn't PCR. There is also reference to TAC. I'm not sure what this is? Perhaps it should be defined in the footnotes? In addition, for some references e.g. 22 the results from hosts are shown in the PCR column.

Figure Files:

Data Requirements:

Reproducibility:

References

---

## [Decision Letter · Decision Letter 2]

16 Oct 2021

Dear Mr Motto,

We are pleased to inform you that your manuscript 'Epidemiology of leptospirosis in Tanzania: A review of the current status, serogroup diversity and reservoirs' has been provisionally accepted for publication in PLOS Neglected Tropical Diseases.

Best regards,

Alan J A McBride, Ph.D.

Associate Editor

Melissa Caimano

Deputy Editor

Reviewer's Responses to Questions

**Key Review Criteria Required for Acceptance?**

**Methods**

-Are the objectives of the study clearly articulated with a clear testable hypothesis stated?

-Is the study design appropriate to address the stated objectives?

-Is the population clearly described and appropriate for the hypothesis being tested?

-Is the sample size sufficient to ensure adequate power to address the hypothesis being tested?

-Were correct statistical analysis used to support conclusions?

-Are there concerns about ethical or regulatory requirements being met?

Reviewer #1: THe authors have addressed my previous comments and I have no additional concerns.

Reviewer #3: (No Response)

**Results**

-Does the analysis presented match the analysis plan?

-Are the results clearly and completely presented?

-Are the figures (Tables, Images) of sufficient quality for clarity?

Reviewer #1: Previous comments have been addressed, and I have no additional comments

Reviewer #3: (No Response)

**Conclusions**

-Are the conclusions supported by the data presented?

-Are the limitations of analysis clearly described?

-Do the authors discuss how these data can be helpful to advance our understanding of the topic under study?

-Is public health relevance addressed?

Reviewer #1: Conclusions are appropriate to the data presented. My previous queries have been addressed and I have no further comments.

Reviewer #3: (No Response)

**Editorial and Data Presentation Modifications?**

Reviewer #1: Nil

Reviewer #3: (No Response)

**Summary and General Comments**

Reviewer #1: All critiques have been addressed. I have no further comments.

Reviewer #3: Very minor comment - In the abstract MAT is defined as "Microagglutination test". It should be "Microscopic Agglutination Test" as defined in the rest of the manuscript.

PLOS authors have the option to publish the peer review history of their article (what does this mean?). If published, this will include your full peer review and any attached files.

Reviewer #1: No

Reviewer #3: **Yes: **Mark Moseley

---

## [Editor Report · Acceptance letter]

11 Nov 2021

Dear Mr Motto,

We are delighted to inform you that your manuscript, "Epidemiology of leptospirosis in Tanzania: A review of the current status, serogroup diversity and reservoirs," has been formally accepted for publication in PLOS Neglected Tropical Diseases.

Best regards,

Shaden Kamhawi

co-Editor-in-Chief

Paul Brindley

co-Editor-in-Chief
